# Data-Driven Distributionally Robust Optimization-Based Coordinated Dispatching for Cascaded Hydro-PV-PSH Combined System

**Shuai Zhang** [1,2], **Gao Qiu** [1,*], **Youbo Liu** [1], **Lijie Ding** [3] **and Yue Shui** [4]

1    College of Electrical Engineering, Sichuan University, Chengdu 610065, China;
     zhangs021x@sc.sgcc.com.cn (S.Z.); liuyoubo@scu.edu.cn (Y.L.)
2    State Grid Chengdu Power Supply Company, Chengdu 610041, China
3    State Grid Sichuan Electric Power Research Institute, Chengdu 610000, China; ding_lijie@163.com
4    Skill Training Center of Sichuan Electric Power Corporation of State Grid, Chengdu 610000, China;
     shuiy0324@sc.sgcc.com.cn
*    Correspondence: qiugscu@scu.edu.cn

**Abstract:** The increasing penetration of photovoltaic (PV) and hydroelectric power generation and their coupling uncertainties have brought great challenges to multi-energy's coordinated dispatch. Traditional methods such as stochastic optimization (SO) and robust optimization (RO) are not feasible due to the unavailability of accurate probability density function (PDF) and over-conservative decisions. This limits the operational efficiency of the clean energies in cascaded hydropower and PV-enriched areas. Based on data-driven distributionally robust optimization (DRO) theory, this paper tailors a joint optimization dispatching method for a cascaded hydro-PV-pumped storage combined system. Firstly, a two-step model for a Distributed Renewable Optimization (DRO) dispatch is developed to create the daily dispatch plan, taking into account the system's complementary economic dispatch cost. Furthermore, the inclusion of a complementary norm constraint is implemented to restrict the confidence set of the probability distribution. This aims to identify the optimal adjustment scheme for the day-ahead dispatch schedule, considering the adjustment cost associated with real-time operations under the most unfavorable distribution conditions. Utilizing the MPSP framework, the Column and Constraint Generation (CCG) algorithm is employed to resolve the two-stage dispatch model. The optimal dispatch schedule is then produced by integrating the daily dispatch plan with the adjustive dispatch scheme. Finally, the numerical dispatch results obtained from an actual demonstration area substantiate the effectiveness and efficiency of the proposed methodology.

**Keywords:** data-driven DRO; cascaded hydro-PV-PSH system; pumped storage hydropower; coordinated dispatch; hydro-PV complementation; C&CG algorithm





## 1. Introduction

The integration of large-scale wind power and PV into the power grid will bring many problems to the power system, such as a sudden drop in regulation capacity, lack of anti-interference ability, and increased risk of chain failure. In order to adapt to the expanding penetration rate of renewable energy and weaken the impact brought by its large-scale integration into the grid, research and exploration work is carried out on the complementary optimization methods of various renewables complementary generation dispatching. It has great practical engineering application value and far-reaching scientific research value. When it comes to the coordinated dispatching of the complementary generation system with multi-dimensional uncertainties containing wind, solar energy, hydro resources and other renewable energies, stochastic optimization (SO) [1] and robust optimization (RO) [2,3] methods are commonly adopted to the model and process. Both domestic and foreign investigators conducted extensive research on this subject, which

contributed to the development of several useful optimization methods, including multi-stage robust optimization [4–6], scenario-based method [7], scenario tree method [8,9], and chance constrained method [10,11]. By extracting scenarios according to the probability distribution of random variables, SO discretizes the probability distribution and generates a large number of discrete samples. It solves each discrete scenario as a deterministic optimization problem separately [9]. The dispatching process of renewable energy generation with multiple uncertainties can be quantitatively analyzed using SO. However, because it is difficult to accurately describe the probability distribution law of the multiple uncertainties in a hybrid renewable power system [12], SO needs to pre-set the probability distribution type [13], which reduces its reliability to a certain extent. Due to the vast number of discrete scenarios that form the basis of SO, calculations will be performed on an unnecessarily large scale, resulting in a longer consuming time and a low calculation efficiency. In addition, although the scenario reduction technology [14] and Benders decomposition acceleration method [15] for discrete scenario can reduce the calculation scale, they cannot cover all the actual scenarios, and the representativeness and typicality of the obtained scenarios are doubtful, then the accuracy of the solution will also be reduced. RO does not need to pre-set the probability distribution of random variables. It describes the random variation characteristics through the boundary parameters of the uncertain variables [16], and a feasible solution can be obtained from the RO model as long as the variable value is within the boundary [17]. Compared with SO, RO can obtain feasible solutions for the uncertain variables that take any value within its boundary, and can strictly ensure the reliability of the decision-making. Besides, the calculation scale is greatly reduced, and the data demand is reduced. One of the deficiencies of RO is that its optimized results tend to be conservative, for that is based on the worst scenario to search the optimal value [3,18]. The chance-constrained method uses the probability index to measure the risk brought by uncertainties in the optimization problem. Ensuring the strict validity of constraints across all practical scenarios is not obligatory, and is receivable that the constraints are valid at a certain confidence level, which will limit the risk caused by the uncertainties to an acceptable range [10,11,17] and seek out the optimal solution on this basis. Different from RO, the chance-constrained method allows for the specification of a desired level of confidence or probability of satisfying constraints [19]. This can reduce the computational overhead associated with considering a vast number of scenarios. The computational effort of chance-constrained methods can be concentrated on relevant confident parts of the uncertainty space. However, its optimization model is generally non-convex and difficult to solve for random variables with complex probability distributions [20]. Approximate solution algorithms [21] and intelligent algorithms [22,23] are used to solve the non-convex chance-constrained models in existing studies, but these algorithms generally cause some problems, such as solution accuracy or calculation efficiency [20], that cannot meet the standards.

To address the challenges associated with the SO, RO, and chance-constrained methods used to manage the uncertain characteristics of renewable energy sources like solar and hydropower, relevant researchers have attempted to integrate SO and RO to create complementary benefits while avoiding the disadvantages of each method separately.

In addition, since power system measuring technology has advanced continuously, an enormous amount of multi-type production data have been generated. Addressing the limited precision of the SO model and the overly conservative nature of the RO model in this context, the emerging need is fulfilled by the innovative approach of data-driven distributionally robust optimization (DRO) [24]. At present, DRO technology has been initially applied in unit combination of power system [25], multi-energy complementation [26], etc., and there are is related research on simplifying the complex calculation process of DRO [27]. Differing from both SO and RO, DRO eliminates the necessity of obtaining the exact probability distribution of variables. Instead, it only necessitates constructing an uncertain set that encompasses the actual distribution and making decisions based on the worst-case distribution. This approach circumvents the challenge of dealing with the

complexity associated with obtaining the exact probability distribution of variables. In addition, techniques like linear decision rules [25] and Lagrangian dual processing [28] can be employed to convert DRO problems into deterministic optimization problems. This transformation helps address the issues of low computational efficiency associated with SO, extensive sampling scale, and the chance-constrained method. A notable advantage of DRO lies in its ability to encompass the probability statistical information pertaining to uncertain parameters, thereby enhancing decision-making by reducing conservativeness. DRO integrates not only the probabilistic statistical features of SO, but also adopts the concept of RO. The decision outcomes exhibit anti-risk capabilities, offering distinct and significant advantages in managing the uncertainties inherent in power systems [29].

In view of the unique advantages of data-driven DRO applied in the fields of uncertain economic dispatching [30], low-carbon dispatching [31], and unit combination [25], this paper suggests a coordinated dispatch approach for a combined cascaded hydro-PV-pumped storage hydropower (CHPP) system, utilizing a data-driven distributionally robust optimization (DRO) method. Initially, the approach constructs a DRO dispatch model consisting of two stages using data-driven methods. The initial phase of the model involves formulating a daily dispatch schedule for the CHPP system, considering the associated economic dispatch costs. Moving on to the subsequent stage, the goal is to confine the confidence set of the probability distribution related to uncertain outputs originating from solar or hydro sources. This stage aims to pinpoint the optimal solution within the context of the most adverse distribution. Considering the adjustment cost linked to real-time system operation, a scheme is formulated to modify the day-ahead dispatch schedule. Finally, the model establishes a coordinated dispatch schedule for the CHPP system. Building upon this foundation, the implementation of the Column and Constraint Generation (CCG) algorithm is undertaken to solve the two-stage DRO dispatch model. Using the actual operation data of the demonstration area to carry out example verification, the outcome indicates that the suggested coordinated dispatch approach adeptly considers the multifaceted uncertainties associated with the CHPP, swiftly achieving an economically optimized dispatch schedule which provides an efficient and practical approach for the uncertain coordinated dispatching of the multi-renewables combined complementary generation system.

## 2. Data-Driven DRO-Coordinated Dispatch Model

The structure and operational mode of the CHPP hybrid system is presented in Figure 1. Balancing the local load is the main goal of CHPP, and the operation scenarios, such as selling electricity to grid and purchasing electricity from grid to pump water for energy storage purpose, are not taken into account. The cascaded hydropower system has an inter-stage spatiotemporal coupling relationship. Outputs of the PV and cascaded system can be used to balance local load and can also be used to pump water. The pumped storage hydropower (PSH) system and the cascaded hydropower system share the upper and lower reservoirs, and there exists an output coupling relationship between them.

### 2.1. Objective Function

When conducting the coordinated dispatching of the CHPP system, minimality of the sum of the day-ahead economic dispatch cost and the real-time output adjustment cost was regarded as the objective function of the DRO model. In the objective function, the economic dispatch cost is composed of the electricity purchase cost from grid, the operation cost of the cascaded hydropower system and PSH system and PV station, which is seen in (1)~(5):

$$C^{daychead} = \sum_{t=1}^{T} \left( C_t^{buy} + C_t^{H} + C_t^{PV} + C_t^{PS} \right), \tag{1}$$

$$C_t^{buy} = \sum_{i=1}^{\Omega} a_t P_{i,t} \cdot \Delta t, \tag{2}$$

$$C_t^{H} = \sum_{i=1}^{N^H} \left[ a_t^H P_{i,t}^H + \delta^H \left( P_{i,t}^{H0} - P_{i,t}^{Hmax} \right) \right] \cdot \Delta t, \tag{3}$$

$$C_t^{PV} = \sum_{i=1}^{N^{PV}} \left[ a_t^{PV} P_{i,t}^{PV} + \delta^{PV} \left( P_{i,t}^{PV0} - P_{i,t}^{PV} \right) \right] \cdot \Delta t, \tag{4}$$

$$C_t^{PS} = \sum_{i=1}^{N^{PS}} \left( \alpha_t^P P_{i,t}^P z_{i,t}^P + \alpha_t^G P_{i,t}^G z_{i,t}^G \right) \cdot \Delta t, \tag{5}$$

where $C^{dayahead}$, $C_t^{buy}$, $C_t^H$, $C_t^{PV}$ and $C_t^{PS}$ are total cost of day-ahead economic dispatching, electricity purchase cost in period $t$, operation cost of the cascaded hydropower system and PV station, and PSH system, respectively. During the $t$ period, $\Omega$ signifies the state parameter related to purchasing electricity, while $N^H$, $N^{PV}$ and $N^{PS}$ represent the quantities of cascaded hydropower facilities, PV facilities, and Pumped Storage Hydropower (PSH) facilities, respectively. Furthermore, $a_t$, $a_t^H$, $a_t^{PV}$, $a_t^P$ and $a_t^G$ denote the cost factors linked to electricity procurement, cascaded hydropower generation, PV generation, PSH water pumping, and PSH generation, respectively. The variable $P_{i,t}$ denotes the purchased power for the $ii$-th state during period $t$. Moreover, $P_{i,t}^H$, $P_{i,t}^{H0}$ and $P_{i,t}^{Hmax}$ denote the output dispatch, day-ahead predicted output value, and maximum allowable output of the $ii$-th stage cascaded hydropower facility in period $t$, respectively. For PV facilities, $P_{i,t}^{PV}$ and $P_{i,t}^{PV0}$ represent the output dispatch and day-ahead predicted output value during period $t$, respectively. Additionally, $P_{i,t}^P$ and $P_{i,t}^G$ indicate the average pumping power and average output of the $ii$-th PSH facility in period $t$, respectively. Additionally, $\delta^H$ and $\delta^{PV}$ function as penalty factors for water curtailment and PV energy curtailment in the day-ahead market. Lastly, $z_{i,t}^P$ and $z_{i,t}^G$ represent the pumping and generation state variables of the $ii$-th stage cascaded hydropower facility in period $t$. These variables indicate pumping water and generation when set to 1; otherwise, they are assigned a value of 0.

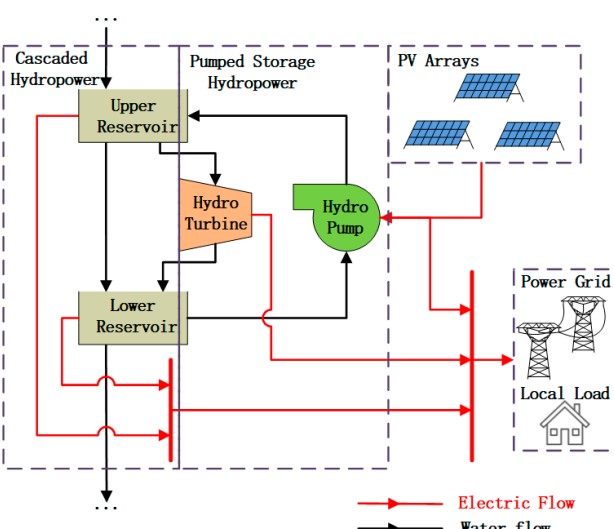

**Figure 1.** Structure and operational mode of a CHPP hybrid power system.

The real-time adjustment cost of the coordinated dispatching of the CHPP complementary system is composed of electricity purchase adjustment cost, cascaded hydropower generation adjustment cost, PV power generation adjustment cost, and PSH operation adjustment cost, which is seen in Formulas (6)–(10):

$$C^{re} = \sum_{t=1}^{T} \left( C_t^{buy\_re} + C_t^{H-re} + C_t^{PV-re} + C_t^{PS-re} \right), \tag{6}$$

$$C_t^{buy\_re} = \sum_{i=1}^{\Omega} c_t \Delta P_{i,t,k} \Delta t, \tag{7}$$

$$C_t^{H-re} = \sum_{i=1}^{N^H} c_{i,t}^H \Delta P_{i,t,k}^H \Delta t, \tag{8}$$

$$C_t^{PV-re} = \sum_{i=1}^{N^{PV}} c_{i,t}^{PV} \Delta P_{i,t,k}^{PV} \Delta t, \tag{9}$$

$$C_t^{PS-re} \& = \sum_{i=1}^{N^{PS}} \left( c_t^P \Delta P_{i,t,k}^P z_{i,t,k}^P + c_t^G \Delta P_{i,t,k}^G z_{i,t,k}^G \right) \Delta t, \tag{10}$$

where $C^{re}$, $C_t^{buy\_re}$, $C_t^{H\_re}$, $C_t^{PV\_re}$ and $C_t^{PS\_re}$ are real-time operation adjustment cost, electricity purchase adjustment cost, operation adjustment cost of cascaded hydropower, PV power station, and PSH station, respectively.

$c_t$, $c_t^H$, $c_t^{PV}$, $c_t^P$ and $c_t^G$ denote the factors influencing the adjustment cost related to purchasing electricity, cascaded hydropower generation, PV power generation, pumping water, and generation of the Pumped Storage Hydropower (PSH) station, respectively. Additionally, $\Delta P_{i,t,k}$, $\Delta P_{i,t,k}^H$, $\Delta P_{i,t,k}^{PV}$, $\Delta P_{i,t,k}^P$ and $\Delta P_{i,t,k}^G$ correspond to the adjustment power associated with electricity purchase for the *kk*-th scenario in period *tt*, the adjustment power of the *ii*-th cascaded hydropower generation, the adjustment power of the *ii*-th PV station, the average adjustment pumping power of the *ii*-th PSH station, and the adjustment generation power of the *ii*-th PSH station, respectively. $z_{i,t,k}^P$ and $z_{i,t,k}^G$ signify the pumping and generation state variables of the *i*-th stage cascaded hydropower station in period *t* for the *k*-th scenario, respectively. These variables indicate the pumping water and generation when set to 1, while being assigned a value of 0 otherwise.

Within the CHPP system, the cost associated with economic dispatch in the day-ahead and the adjustment cost for real-time operation are both considered. The objective function of the DRO-coordinated dispatch model is established by minimizing the combined cost of these two aspects, as outlined in the Equation (11):

$$min \left\{ C^{\, dayahead} + \max_{\{p_k\}} \left[ \sum_{k=1}^K p_k min(C^{re}) \right] \right\}, \tag{11}$$

Formula (11) introduces a three-layered two-stage robust optimization problem involving min-max-min. In contrast to traditional two-stage robust optimization, where the focus is solely on optimizing for the worst-case scenario, the internal max-min function in the DRO model calculates the most unfavorable probability distribution across *K* discrete scenarios by optimizing decision variables $\{p_k\}$. It then determines the maximum expected cost value, which is utilized as a comprehensive adjustment cost for real-time CHPP operation and is incorporated into the objective function.

*2.2. Constraints*

The constraints of the two-stage DRO model for CHPP are mainly divided into three types: day-ahead dispatch constraints, real-time dispatch constraints, and data-driven comprehensive norm constraints. These constraints will be discussed separately below.

2.2.1. Day-Ahead Dispatch Constraints

The day-ahead dispatch constraints are formed based on the forecasting information of cascaded hydropower system and PV system, and the constraints mainly include power balance constraint, operation constraints of CHPP, grid constraint and reserve constraints, etc.

1.  Power Balance Constraint

$$\sum_{i=1}^{\Omega} P_{i,t} + \sum_{i=1}^{N^H} P_{i,t}^H + \sum_{i=1}^{N^{PV}} P_{i,t}^{PV} - \sum_{i=1}^{N^{PS}} P_{i,t}^P z_{i,t}^P + \sum_{i=1}^{N^{PS}} P_{i,t}^G z_{i,t}^G = \sum_{i=1}^{\Omega} P_{i,t}^L, \tag{12}$$

where $P_{i,t}^L$ is the system load power of each state in period *t*.

2.  Operation Constraints of the Cascaded Hydropower
    a.  Water Storage Variation Constraint of Reservoir

$$V_i^{min} \leq V_{i,t} \leq V_i^{max}, \tag{13}$$

where, in period *t*, $V_{i,t}$ signifies the reservoir's water capacity, while $V_i^{max}$ and $V_i^{min}$ represent the maximum and minimum capacity of the *ii*-th reservoir, respectively.

b.  Output Constraint

$$P_i^{Hmin} \leq P_{i,t}^H \leq P_{i,t}^{H\circ}, \tag{14}$$

where $P_i^{Hmin}$ denotes the lowest acceptable output value for the $i$-th cascaded hydropower station.

c.  Discharging Flow Constraint

$$Q_i^{Hmin} \leq Q_{i,t}^H \leq Q_i^{Hmax}, \tag{15}$$

where $Q_{i,t}^H$ denotes the outflow rate from the $i$-th reservoir during a specific period. $Q_i^{Hmax}$ and $Q_i^{Hmin}$ represent the maximum and minimum outflow rates.

d.  Water Volume Balance Constraint

$$V_{i,t+1} = V_{i,t} + \left( I_{i,t} - Q_{i,t}^H \right) \Delta t = V_{i,t} + \left( I_{i,t} - Q_{i,t}^G - Q_{i,t}^C \right) \Delta t, \tag{16}$$

where $I_{i,t}$, $Q_{i,t}^G$ and $Q_{i,t}^C$ are reservoir inflow, flow of power generation, and outflow from the $i$-th reservoir.

e.  Constraint related to hydraulic connectivity between stages

$$I_{i+1,t+\tau} = Q_{i,t}^H + L_{i,t}, \tag{17}$$

where $\tau$ and $L_{i,t}$ represent the delay factors associated with the water flow between stages and the interval inflow between stages.

f.  Limit on the rate of change in output power for the hydraulic turbine

$$\delta^L \Delta t \leq \left( P_{i,t+1}^H - P_{i,t}^H \right) \leq \delta^U \Delta t, \tag{18}$$

where $\delta^L$ and $\delta^U$ denote the minimum and maximum rates of change in output for the cascaded hydropower system.

3.  Constraints on the Output of PV Station

$$0 \leq P_{i,t}^{PV} \leq P_{i,t}^{PV0}, \tag{19}$$

4.  Operation Constraints of PSH

a.  Restriction on the Fluctuation of Reservoir Water Storage

$$V_i^{Umin} \leq V_{i,t}^U \leq V_i^{Umax}, \tag{20}$$

$$V_i^{Lmin} \leq V_{i,t}^L \leq V_i^{Lmax}, \tag{21}$$

where $V_{i,t}^U$, $V_i^{Umax}$ and $V_i^{Umin}$ are the upper reservoir's water storage volume in period $t$, and the maximum and minimum reservoir capacities of the $i$-th PSH station, respectively; $V_{i,t}^L$, $V_i^{Lmax}$, $V_i^{Lmin}$ are the lower reservoir's water storage volume in period $t$, and the maximum and minimum reservoir capacities of the $i$-th PSH station, respectively.

b.  Water Volume Balance Constraint Constraint in Generation State

$$V_{i,t+1}^U = V_{i,t}^U - \Delta t \frac{P_{i,t}^G}{\eta_G}, \tag{22}$$

$$V_{i,t+1}^L = V_{i,t}^L + \Delta t \frac{P_{i,t}^G}{\eta_G}, \tag{23}$$

Constraint in Pumping State

$$V_{i,t+1}^U = V_{i,t}^U + \Delta t \eta_P P_{i,t}^P, \tag{24}$$

$$V_{i,t+1}^L = V_{i,t}^L - \Delta t \eta_P P_{i,t}^P, \tag{25}$$

where $\eta_G$ and $\eta_P$ are the generation efficiency and pumping efficiency of PSH, respectively.

c. Constraints of Generation Power and Pumping Power

$$P_i^{Gmin} \le P_{i,t}^G \le min\left( P_i^{Gmax}, \frac{V_{i,t}^U}{\Delta t}\eta_G \right) \text{or} P_{i,t}^G = 0, \tag{26}$$

$$P_i^{Pmin} \le P_{i,t}^P \le min\left( P_i^{Pmax}, \frac{V_{i,t}^L}{\Delta t \eta_P} \right) \text{or} P_{i,t}^P = 0, \tag{27}$$

where $P_i^{Gmin}$ and $P_i^{Gmax}$ are the minimum and maximum generation power of the $i$th PSH station, respectively. $P_i^{Pmin}$ and $P_i^{Pmax}$ are the minimum and maximum pumping power of the $i$th PSH station, respectively.

d. Electricity Balance Constraint of Generation and Pumping

$$\sum_{t=1}^T P_{i,t}^G \Delta t = \eta_P \eta_G \sum_{t=1}^T P_{i,t}^P \Delta t, \tag{28}$$

e. Ramp Rate Constraint of PSH

$$\delta_G^L \Delta t \le \left( P_{i,t+1}^G - P_{i,t}^G \right) \le \delta_G^U \Delta t, \tag{29}$$

$$\delta_P^L \Delta t \le \left( P_{i,t+1}^P - P_{i,t}^P \right) \le \delta_P^U \Delta t, \tag{30}$$

where $\delta_G^L$ and $\delta_G^U$ are the minimum and maximum variation speed of generation power, and $\delta_P^L$ and $\delta_P^U$ are the minimum and maximum variation speed of pumping power, respectively.

f. Operating State Mutual Exclusion Constraint

$$z_{i,t}^G \cdot z_{i,t}^P = 0, \tag{31}$$

5. Grid Flow Constraint
The DC power flow constraints in the literature [32] are taken as the grid flow constraint of the hybrid system. Each branch flow should meet the following (32):

$$\begin{cases} P_{tline} = B_{diag}LB^{-1}\left( P_t + P_t^H + P_t^{PV} - P_t^P z_t^P + P_t^G z_t^G - P_t^L \right) \\ -\overline{P}_{line} \le P_{tline} \le \overline{P}_{line} \\ B_{diag} = diag\left( \frac{1}{x_1}, \frac{1}{x_2}, \cdots, \frac{1}{x_N} \right) \end{cases} \tag{32}$$

For each branch, $P_{tline}$ represents the DC power flow, with $B$, $L$ as the admittance coefficient matrix and the node connection matrix of the network branch. Furthermore, $P_t$, $P_t^H$, $P_t^{PV}$, $P_t^P$, $P_t^G$ and $P_t^L$ serve as vector representations for the purchased power, cascaded system output, PV system output, PSH system pumping power, PSH system generation power, and load demand during period $tttt$, respectively. $z_t^P$ and $z_t^G$ denote the PSH pumping state vector and generation state vector during period $t$, respectively. $\overline{P}_{line}$ stands for the maximum branch power, $N$ is the reactance of the $ll$-th branch, and $zz$ indicates the branch number in the network.

6. Reserve Constraints

The complementary generation system's requirements for upward and downward reserve capacity are fulfilled by the cascaded hydropower system and the PSH system. The system's reserve requirement is calculated as the sum of the partial output from PV and the partial output from the cascaded hydropower, as specified in the Equations (33) and (34):

$$\sum_{i=1}^{N^H}\left(P_{i,t}^{H0}-P_{i,t}^{H}\right)+\sum_{i=1}^{N^{PS}}\left(min\left[P_i^{Gmax},\frac{V_{i,t}^U}{\Delta t}\eta_G\right]-P_{i,t}^{G}z_{i,t}^{G}\right)$$
$$\geq \gamma^{PV}\sum_{i=1}^{N^H}P_{i,t}^{PV}+\gamma^{H}\sum_{i=1}^{N^H}P_{i,t}^{H}, \tag{33}$$

$$\sum_{i=1}^{N^H}\left(P_{i,t}^{H}-P_{i,t}^{Hmin}\right)+\sum_{i=1}^{N^{PS}}\left(P_{i,t}^{G}z_{i,t}^{G}-P_{i,t}^{Gmin}\right)$$
$$\geq \gamma^{PV}\sum_{i=1}^{N^H}P_{i,t}^{PV}+\gamma^{H}\sum_{i=1}^{N^H}P_{i,t}^{H}, \tag{34}$$

where $\gamma^{PV}$ represents the spinning reserve factor for PV output, set at 10% in this context, and $\gamma^{H}$ is the spinning reserve factor for the cascaded hydropower system, set at 4%.

### 2.2.2. Real-Time Dispatch Constraints

Due to the uncertainties of the CHPP hybrid system in its real-time operation stage, it needs to adjust the output of cascaded hydropower and PSH to regulate the random real-time fluctuations of hydro and solar output. The real-time dispatch constraints include power balance constraint, real-time operation constraints of cascaded hydropower, PV and PSH system, grid flow constraint and reserve constraints, etc.

1. Regulated Power Balance Constraint

$$\sum_{i=1}^{\Omega}(P_{i,t}+\Delta P_{i,t,k})+\sum_{i=1}^{N^H}\left(P_{i,t}^{H}+\Delta P_{i,t,k}^{H}\right)+\sum_{i=1}^{N^{PV}}\left(P_{i,t}^{PV}+\Delta P_{i,t,k}^{PV}\right)-$$
$$\sum_{i=1}^{N^{PS}}\left[\left(P_{i,t}^{P}+\Delta P_{i,t,k}^{P}\right)z_{i,t}^{P}\right]+\sum_{i=1}^{N^{PS}}\left[\left(P_{i,t}^{G}+\Delta P_{i,t,k}^{G}\right)z_{i,t}^{G}\right]=\sum_{i=1}^{\Omega}P_{i,t}^{L}, \tag{35}$$

2. Operation Constraints of the Cascaded Hydropower

   a. Water Storage Regulated Variation Constraint of Reservoir

$$V_i^{min}\leq(V_{i,t}+\Delta V_{i,t,k})\leq V_i^{max}, \tag{36}$$

   where $\Delta V_{i,t,k}$ is the water storage regulated variation of the *i*th reservoir under the *k*th scenario in period *t*.

   b. Regulated Output Constraint

$$P_i^{Hmin}\leq\left(P_{i,t}^{H}+\Delta P_{i,t,k}^{H}\right)\leq P_{i,t}^{H0}, \tag{37}$$

   c. Regulated Discharging Flow Constraint

$$Q_i^{Hmin}\leq\left(Q_{i,t}^{H}+\Delta Q_{i,t,k}^{H}\right)\leq Q_i^{Hmax}, \tag{38}$$

   where $\Delta Q_{i,t,k}^{H}$ is the regulated discharging flow of the *i*th reservoir under the *k*th scenario in period *t*.

d.  Regulated Water Volume Balance Constraint

$$
\begin{aligned}
V_{i,t+1} + \Delta V_{i,t+1,k} &= V_{i,t} + \Delta V_{i,t,k} + \left( I_{i,t} + \Delta I_{i,t,k} - Q_{i,t}^H - \Delta Q_{i,t,k}^H \right) \Delta t \\
&= V_{i,t} + \Delta V_{i,t,k} + \begin{pmatrix} I_{i,t} + \Delta I_{i,t,k} - Q_{i,t}^G - \Delta Q_{i,t,k}^G \\ -Q_{i,t}^C - \Delta Q_{i,t,k}^C \end{pmatrix} \Delta t
\end{aligned}
\tag{39}
$$

where $\Delta I_{i,t,k}$, $\Delta Q_{i,t,k}^G$ and $\Delta Q_{i,t,k}^C$ are the regulated inflow, regulated generation flow, and regulated discharging flow of the $i$th reservoir under $k$th scenario in period $t$, respectively.

e.  Interstage Hydraulic Connection Constraint

$$
I_{i+1,t+\tau} + \Delta I_{i+1,t+\tau,k} = Q_{i,t}^H + \Delta Q_{i,t,k}^H + L_{i,t} + \Delta L_{i,t,k},
\tag{40}
$$

f.  Ramp Rate Constraint of Hydraulic Turbine

$$
\delta^L \Delta t \le \left( P_{i,t+1}^H + \Delta P_{i,t+1,k}^H - P_{i,t}^H - \Delta P_{i,t,k}^H \right) \le \delta^U \Delta t,
\tag{41}
$$

3.  Output Constraints of PV station

$$
0 \le \left( P_{i,t}^{PV} + \Delta P_{i,t,k}^{PV} \right) \le P_{i,t}^{PV0},
\tag{42}
$$

4.  Operation Constraints of PSH

a.  Water Storage Regulated Variation Constraint of Reservoir

$$
V_i^{Umin} \le \left( V_{i,t}^U + \Delta V_{i,t,k}^U \right) \le V_i^{Umax},
\tag{43}
$$

$$
V_i^{Lmin} \le \left( V_{i,t}^L + \Delta V_{i,t,k}^L \right) \le V_i^{Lmax},
\tag{44}
$$

where $\Delta V_{i,t,k}^U$ and $\Delta V_{i,t,k}^L$ are upper reservoir regulated water storage and lower reservoir regulated water storage of the $i$th PSH station in period $t$, respectively.

b.  Regulated Water Volume Balance Constraint Constraint in Generation State

$$
V_{i,t+1}^U + \Delta V_{i,t+1,k}^U = V_{i,t}^U + \Delta V_{i,t,k}^U - \Delta t \frac{P_{i,t}^G + \Delta P_{i,t,k}^G}{\eta_G},
\tag{45}
$$

$$
V_{i,t+1}^L + \Delta V_{i,t+1,k}^L = V_{i,t}^L + \Delta V_{i,t,k}^L + \Delta t \frac{P_{i,t}^G + \Delta P_{i,t,k}^G}{\eta_G},
\tag{46}
$$

Constraint in Pumping State

$$
V_{i,t+1}^U + \Delta V_{i,t+1,k}^U = V_{i,t}^U + \Delta V_{i,t,k}^U + \Delta t \eta_P \left( P_{i,t}^P + \Delta P_{i,t,k}^P \right),
\tag{47}
$$

$$
V_{i,t+1}^L + \Delta V_{i,t+1,k}^L = V_{i,t}^L + \Delta V_{i,t,k}^L - \Delta t \eta_P \left( P_{i,t}^P + \Delta P_{i,t,k}^P \right),
\tag{48}
$$

c.  Constraints of Regulated Generation Power and Regulated Pumping Power

$$
P_i^{Gmin} \le \left( P_{i,t}^G + \Delta P_{i,t,k}^G \right) \le min \left( P_i^{Gmax}, \frac{V_{i,t}^U + \Delta V_{i,t,k}^U}{\Delta t} \eta_G \right) \text{or } P_{i,t}^G = 0,
\tag{49}
$$

$$
P_i^{Pmin} \le \left( P_{i,t}^P + \Delta P_{i,t,k}^P \right) \le min \left( P_i^{Pmax}, \frac{V_{i,t}^L + \Delta V_{i,t,k}^L}{\Delta t \eta_P} \right) \text{or } P_{i,t}^P = 0,
\tag{50}
$$

d.　　Regulated Electricity Balance Constraint of Generation and Pumping

$$\sum_{t=1}^{T}\left(P_{i,t}^{G}+\Delta P_{i,t,k}^{G}\right)\Delta t \ = \eta_P\eta_G\sum_{t=1}^{T}\left(P_{i,t}^{P}+\Delta P_{i,t,k}^{P}\right)\Delta t, \tag{51}$$

e.　　Ramp Rate Constraint of PSH

$$\delta_G^L\Delta t \ \leq \left(P_{i,t+1}^{G}+\Delta P_{i,t+1,k}^{G} - P_{i,t}^{G}-\Delta P_{i,t,k}^{G}\right)\leq \delta_G^U\Delta t, \tag{52}$$

$$\delta_P^L\Delta t \ \leq \left(P_{i,t+1}^{P}+\Delta P_{i,t+1,k}^{P} - P_{i,t}^{P}-\Delta P_{i,t,k}^{P}\right)\leq \delta_P^U\Delta t, \tag{53}$$

f.　　Operating State Mutual Exclusion Constraint

$$z_{i,t}^{G}\cdot z_{i,t}^{P}=0, \tag{54}$$

5.　　Grid Flow Constraint

$$\begin{cases} P_{tline} = B_{diag}LB^{-1}\left[(P_t+\Delta P_{t,k}) + \left(P_t^{H}+\Delta P_{t,k}^{H}\right)+P_{t,k}^{PV} - \left(P_t^{P}+\Delta P_{t,k}^{P}\right)z_{t,k}^{P} + \left(P_t^{G}+\Delta P_{t,k}^{G}\right)z_{t,k}^{G} - P_t^{L}\right] \\ \qquad -\bar{P}_{line} \leq P_{tline} \leq \bar{P}_{line} \\ B_{diag} = diag\left(\frac{1}{x_1},\frac{1}{x_2},\cdots,\frac{1}{x_N}\right) \end{cases}, \tag{55}$$

where $\Delta P_{t,k}$, $\Delta P_{t,k}^{H}$, $\Delta P_{t,k}^{P}$ and $\Delta P_{t,k}^{G}$ are the vector forms of regulated purchased power, regulated output power of the cascaded system, regulated pumping power of PSH, and regulated generation power of PSH under the *k*th scenario in period *t* respectively.

6.　　Reserve Constraints

$$\sum_{i=1}^{N^H}\left(\begin{array}{c}P_{i,t}^{H0}-P_{i,t}^{H}- \\ \Delta P_{i,t,k}^{H}\end{array}\right) + \sum_{i=1}^{N^{PS}}\left(\begin{array}{c}min\left[P_i^{Gmax},\frac{V_{i,t}^{U}+\Delta V_{i,t,k}^{U}}{\Delta t}\eta_G\right] \\ -\left(P_{i,t}^{G}+\Delta P_{i,t,k}^{G}\right)z_{i,t}^{G}\end{array}\right), \\ \geq \gamma^{PV}\sum_{i=1}^{N^H}\left(P_{i,t}^{PV}+\Delta P_{i,t,k}^{PV}\right)+\gamma^{H}\sum_{i=1}^{N^H}\left(P_{i,t}^{H}+\Delta P_{i,t,k}^{H}\right) \tag{56}$$

$$\sum_{i=1}^{N^H}\left(P_{i,t}^{H}+\Delta P_{i,t,k}^{H}-P_{i,t}^{Hmin}\right) + \sum_{i=1}^{N^{PS}}\left[\begin{array}{c}\left(P_{i,t}^{G}+\Delta P_{i,t,k}^{G}\right)z_{i,t}^{G}- \\ P_{i,t}^{Gmin}\end{array}\right], \\ \geq \gamma^{PV}\sum_{i=1}^{N^H}\left(P_{i,t}^{PV}+\Delta P_{i,t,k}^{PV}\right)+\gamma^{H}\sum_{i=1}^{N^H}\left(P_{i,t}^{H}+\Delta P_{i,t,k}^{H}\right) \tag{57}$$

2.2.3. Data-Driven Comprehensive Norm Constraints

Because the conventional solution method is too complex for solving the DRO-coordinated dispatching model [27], this paper introduced a data-driven DRO algorithm utilizing 1-norm and ∞-norm to solve the model. The algorithm takes the historical data of uncertain parameters, such as hydraulic runoff and solar intensity, as reference, and screens the hydropower and PV output of *K* discrete scenarios, as well as the initial probability of each scenario by extracting hydraulic and solar historical data of limited typical days. Then, centering on each initial probability distribution, the comprehensive norm constraint is introduced to calculate the joint optimization problem so as to derive the most unfavorable probability distribution for each discrete scenario, aiming to attain the maximum expected objective value within this particular setting. Therefore, in the data-driven CHPP two-stage DRO-coordinated dispatching model, in addition to the real-time dispatching constraints and day-ahead dispatching constraints, it is necessary to take the comprehensive norm constraints into consideration.

The comprehensive norm, which includes both the 1-norm and ∞-norm, places constraints on the discrete scenarios of hydro-solar random output. $\Omega$ denotes the region where the comprehensive norm is feasible, and can be expressed as Equation (58):

$$\Omega = \left\{ \{p_k\} \left| \begin{array}{l} p_k \geq 0, \ k = 1, 2, \cdots, K \\ \sum_{k=1}^{K} p_k = 1 \\ \sum_{k=1}^{K} \left| p_k - p_k^0 \right| \leq \theta_1 \\ \max_{1 \leq k \leq K} \left| p_k - p_k^0 \right| \leq \theta_\infty \end{array} \right. \right\}, \tag{58}$$

where $p_k^0$ denotes the initial probability for $k$ discrete scenarios; $\sum_{k=1}^{K} \left| p_k - p_k^0 \right| \leq \theta_1$ is 1-norm constraint and $\max_{1 \leq k \leq K} \left| p_k - p_k^0 \right| \leq \theta_\infty$ is $\infty$-norm constraint; $\theta_1$, $\theta_\infty$ represent the acceptable deviation limits for the probability of discrete scenarios.

The confidence coefficient of $\{p_k\}$ can be defined using Equations (59) and (60), as outlined in references [33,34].

$$\begin{cases} Pr\left\{ \sum_{k=1}^{K} \left| p_k - p_k^0 \right| \leq \theta_1 \right\} \geq \alpha_1 \\ Pr\left\{ \max_{1 \leq k \leq K} \left| p_k - p_k^0 \right| \leq \theta_\infty \right\} \geq \alpha_\infty \end{cases}, \tag{59}$$

$$\begin{cases} \alpha_1 = 1 - 2Ke^{-\frac{2M\theta_1}{K}} \\ \alpha_\infty = 1 - 2Ke^{-2M\theta_\infty} \end{cases}, \tag{60}$$

where $\alpha_1$, $\alpha_\infty$ signify the confidence coefficients for the probability distribution within the set of discrete scenarios under the 1-norm and $\infty$-norm constraints. $M$ represents the count of chosen days with limited typical hydro-solar output.

Based on Formulas (59) and (60), the allowable deviation limits $\theta_1$ and $\theta_\infty$ constraining hydro-solar uncertain outputs can be obtained, as shown in (61):

$$\begin{cases} \theta_1 = \frac{K}{2M} \ln \frac{2K}{1 - \alpha_1} \\ \theta_\infty = \frac{1}{2M} \ln \frac{2K}{1 - \alpha_\infty} \end{cases}, \tag{61}$$

## 3. Solution Algorithm for the Dro Dispatch Model

### 3.1. Linear Treatment for Comprehensive Norm Constraints

The absolute value expression of the comprehensive norm constraints in Formula (58) will bring a lot of in convenience to the solution of the model. The approach in literature [26] is adopted, and 0–1 auxiliary variables are introduced to convert the absolute value constraints to equivalent linear constraints.

For 1-norm constraint, auxiliary variables $x_k^+$ and $x_k^-$ are introduced, and the details are shown in (62) and (63). This constraint can be linearly converted to (64):

$$x_k^+ + x_k^- \leq 1 \ \forall k, \tag{62}$$

$$\begin{cases} 0 \leq p_k^+ \leq x_k^+ \theta_1 \forall k \\ 0 \leq p_k^- \leq x_k^- \theta_1 \forall k \\ p_k = p_k^0 + p_k^+ + p_k^- \forall k \end{cases}, \tag{63}$$

$$\sum_{k=1}^{K} \left( p_k^+ + p_k^- \right) \leq \theta_1, \tag{64}$$

where $p_k^+$ and $p_k^-$ are the positive and negative offset of $p_k$ relative to the initial probability $p_k^0$ respectively.

Similarly, 0–1 auxiliary variables $y_k^+$ and $y_k^-$ shown in Formulas (65) and (66), are introduced to convert the $\infty$-norm constraint to its equivalent linear form, as shown in Formula (67):

$$y_k^+ + y_k^- \leq 1 \forall k, \tag{65}$$

$$
\left\{
\begin{array}{l}
0 \leq p_k^+ \leq y_k^+ \theta_\infty \forall k \\
0 \leq p_k^- \leq y_k^- \theta_\infty \forall k \\
p_k = p_k^0 + p_k^+ + p_k^- \forall k
\end{array}
\right. , \tag{66}
$$

$$
p_k^+ + p_k^- \leq \theta_\infty \forall k, \tag{67}
$$

### 3.2. Model Solution for DRO Coordinated Dispatch

Common methods for two-stage RO model solution include Affine Policy (AP, also called Linear Decision Rule, LDR) [35,36], Benders Decomposition [15], Column and Constraint Generation (CCG) [37], etc. Compared with AP and Benders Decomposition, CCG can achieve a faster solving speed and better convergence characteristics, and shows a better optimality of the solution [17]. The CCG algorithm is employed for computing the data-driven CHPP two-segment DRO-coordinated dispatching model. The first stage of the model is the day-ahead dispatching decision based on decision variables such as the output plan of CHPP under the water and PV forecast information. In the subsequent stage, the objective is to determine the real-time output adjustment for the CHPP system, utilizing the actual data of water and PV in the operational scenario.

The fundamental approach of the CCG algorithm involves breaking down the two-stage DRO model into two components: the sub-problem (SP) and the master problem (MP), which are then repeatedly solved. The Master Problem (MP) seeks an optimal solution under the assumption of a known finite unfavorable probability distribution, thereby establishing the lower limit value for the two-segment DRO model. The MP is represented in Equations (68) and (69).

$$
\min_{x \in X, y_0 \in (x,\xi_0), y_k^{(m)} \in Y(x,\xi_k), L} C^{dayahead}(x, y_0, \xi_0) + L, \tag{68}
$$

$$
L \geq \sum_{k=1}^{K} p_k^{(m)} C^{re}\left(y_k^{(m)}, \xi_k\right), \forall m = 1, 2, \cdots, n, \tag{69}
$$

where $x$ represents the choice variables, $y_k$ is the second stage choice variables in the $k$th scenario, $y_0$ denotes the choice variables for the two-segment based on forecast information, $\xi_0$ is the predicted values of water runoff and light intensity, $\xi_k$ is the predicted values of water runoff and light intensity in the $k$th discrete scenario, and $m$ is the iteration count.

SP finds the probability distribution of the worst scenario under the situation that the choice variables $x$ in the first stage are given $(x^*)$, provides the upper limit value for the model, and brings it to MP for the next iterative calculation. SP can be described by Formula (70). In its inner min-function, since all scenarios are independent of each other, the parallel algorithm can be introduced to accelerate the calculation process. If the inner function is represented by $f(x^*, \xi_k)$, Formula (70) can be changed to Formula (71). The continuous iterations and updates are carried out based on Formulas (68), (69) and (71) until the calculation accuracy of the model reaches predetermined level.

$$
L(x^*) = \max_{\{p_k\} \in \Omega} \sum_{k=1}^{K} p_k \min_{y_k \in Y(x^*, \xi_k)} C^{re}(y_k, \xi_k), \tag{70}
$$

$$
L(x^*) = \max_{\{p_k\} \in \Omega} \sum_{k=1}^{K} p_k \cdot f(x^*, \xi_k), \tag{71}
$$

The calculating process of the data-driven CHPP two-stage DRO coordinated dispatching model is shown in Figure 2.

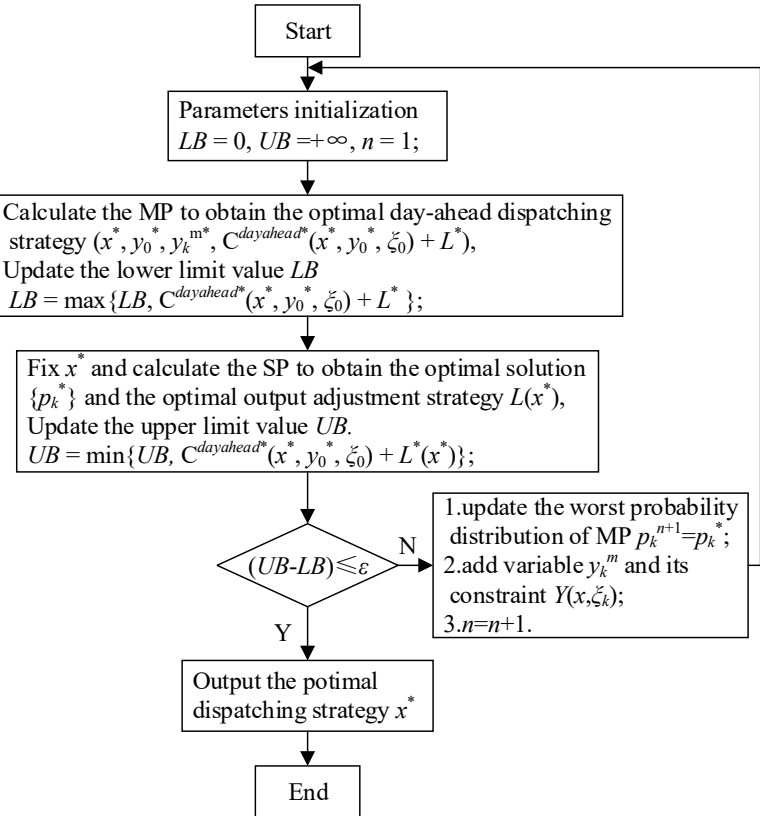

**Figure 2.** Calculating process of the two-stage DRO model. * implies the optimal parameters.

## 4. Numerical Study

The improved 24-node test system, shown in Figure 3 [38], was introduced to test the effectiveness and feasibility of the two-stage DRO-coordinated dispatch method for the CHPP system. There are 11 renewable power stations located at 1#, 2#, 7#, 13#, 14#, 15#, 16#, 18#, 21#, 22#, and 23# node, respectively, and the main parameters of the stations are shown in Table 1. The station located at node 7# belongs to run-off hydropower station, power station without regulating reservoir, whose power generation capacity depends on water flow upriver and lacks generation adjustment ability. The other hydropower stations belong to regulated hydropower station and have daily adjustment ability. Generally, daily regulation of this type of station is usually carried out only in the dry season, and runoff power generation is often used in flood season. The cascaded hydropower station adopts a cascade hydraulic development, which starts from the upper reaches of the river, and a series of water conservancy hubs are drawn up from the top to the bottom in a ladder-like distribution form. A series of hydropower stations built by cascade development are called cascaded hydropower stations. In the test system, cascaded hydropower station 1 (CHS 1) is the leading power station of the entire cascade system, and CHS 2–CHS 7 are downstream stations in sequence. CHS 4 and CHS 5 are connected in parallel with CHS 6 and CHS 7, respectively. Pumped storage hydropower station (PSH) uses the electric energy in the low load to pump water to the upper reservoir, and then releases water to the lower reservoir to generate electricity in the peak load. It can convert the excess energy when the power grid load is low into high-value electric energy during the peak period of the power grid, and is also suitable for frequency and phase regulation to stabilize the frequency and voltage of the power system; it is also suitable for emergency backup. In the 24-node system, the PSH is located at BUS 16.

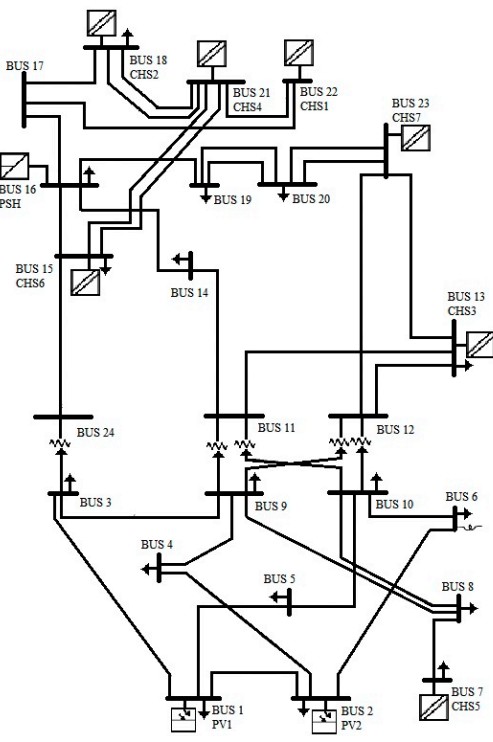

**Figure 3.** Topology for 24-node test system.

**Table 1.** Main parameters of each station.

| Node | Station Type | Installed Capacity /MW | Reservoir Capacity /104 m³ | Power Generation Head/m | Working Flow /m³/s |
|------|--------------|------------------------|----------------------------|-------------------------|---------------------|
| 1# | PV 1 | 50 | - | - | - |
| 2# | PV 2 | 50 | - | - | - |
| 7# | CHS 5 | 20 | - | 339.09 | 7.4 |
| 13# | CHS 3 | 36 | 37.2 | 91 | 33 |
| 14# | - | - | - | - | - |
| 15# | CHS 6 | 4.8 | 53 | 10.5 | 48.9 |
| 16# | PSH | 5 | 25.6/23.7 | 339.09 | 7.4 |
| 18# | CHS 2 | 60 | 20.9 | 127 | 53.4 |
| 21# | CHS 4 | 54 | 25.6 | 161 | 47.1 |
| 22# | CHS 1 | 45 | 16.1 | 123.65 | 43.32 |
| 23# | CHS 7 | 84 | 44.2 | 75 | |

According to the climatic characteristics of typical regions and the characteristics of PV and hydropower output, the sunny and rainy weather type that affects photovoltaic output, and the dry and wet seasons that affect hydropower output, are selected here to form four representative water-light operation scenarios: sunny/wet season, rainy/wet season, sunny/dry season, rainy/dry season. They are considered to carry out the coordinated dispatch applicating research. In each scenario, the forecast value of PV output, the forecast value of the cascaded system's natural runoff output, and the forecast value of local load demand are shown in Table 2.

**Table 2.** Forecast value of PV, cascaded system, and local.

| Time/h | PV Stations | | Cascaded System | | Local Load |
|---|---|---|---|---|---|
| | **Sunny** | **Rainy** | **Wet Season** | **Dry Season** | |
| 1 | 0 | 0 | 290.62 | 160.44 | 134 |
| 2 | 0 | 0 | 292.44 | 153.56 | 134 |
| 3 | 0 | 0 | 275.18 | 161.22 | 133 |
| 4 | 0 | 0 | 283.08 | 151.88 | 133 |
| 5 | 0 | 0 | 290.6 | 156.44 | 132 |
| 6 | 0 | 0 | 287.32 | 152.59 | 132 |
| 7 | 0 | 0 | 297.48 | 153.4 | 134 |
| 8 | 4.53 | 4.86 | 288.06 | 145.71 | 137 |
| 9 | 14.23 | 3.6 | 292.25 | 163.51 | 141 |
| 10 | 28.69 | 14.69 | 269.49 | 147.82 | 140 |
| 11 | 34.57 | 3.97 | 296.45 | 162.09 | 139 |
| 12 | 39.76 | 8.1 | 287.71 | 160.08 | 140 |
| 13 | 40.24 | 8.68 | 293.47 | 149.98 | 138 |
| 14 | 40.03 | 6.24 | 282.16 | 164.76 | 138 |
| 15 | 39.5 | 11.83 | 291.02 | 152.99 | 138 |
| 16 | 30.84 | 6.98 | 282.7 | 148.82 | 138 |
| 17 | 18.15 | 3.58 | 289.05 | 151.72 | 138 |
| 18 | 7.53 | 2.13 | 282.85 | 148.37 | 139 |
| 19 | 0 | 0.1 | 276.96 | 154.21 | 140 |
| 20 | 0 | 0 | 269.63 | 152.03 | 140 |
| 21 | 0 | 0 | 286.37 | 146.95 | 140 |
| 22 | 0 | 0 | 298.45 | 158.84 | 140 |
| 23 | 0 | 0 | 292.73 | 162.34 | 139 |
| 24 | 0 | 0 | 293.8 | 151.65 | 137 |

The confidence levels $\alpha_1$ and $\alpha_\infty$ are set to 0.2 and 0.8, respectively. The number of discrete scenarios, $K$, is 50, and the number of historical data, $M$, is set to 1000. Software CPLEX 12.6 and a computing platform with a 3.10 GHz processor and 4 GB memory were introduced to calculate the solution for the application example. The optimal result of the CHPP-coordinated dispatching in four applied scenarios are shown in Figures 4–7.

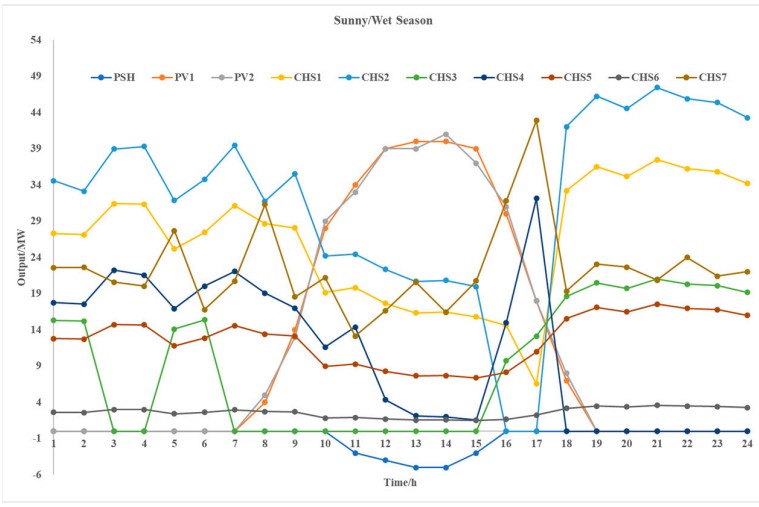

**Figure 4.** Optimal dispatch result in sunny/wet season scenario.

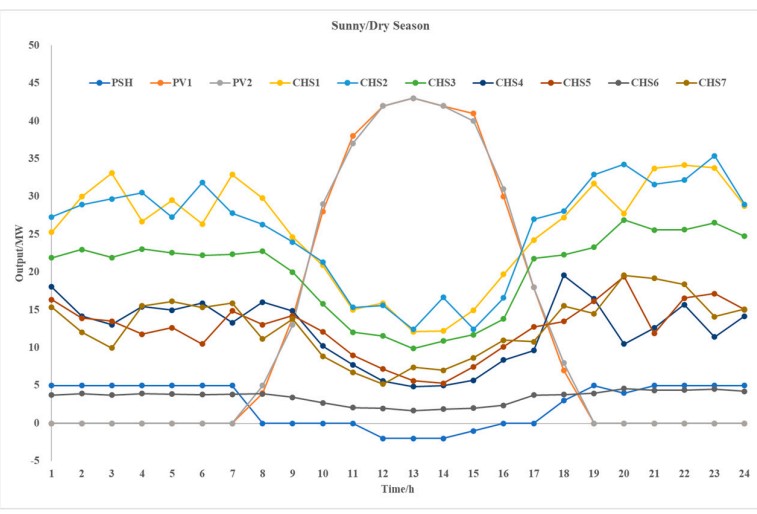

**Figure 5.** Optimal dispatch result in sunny/dry season scenario.

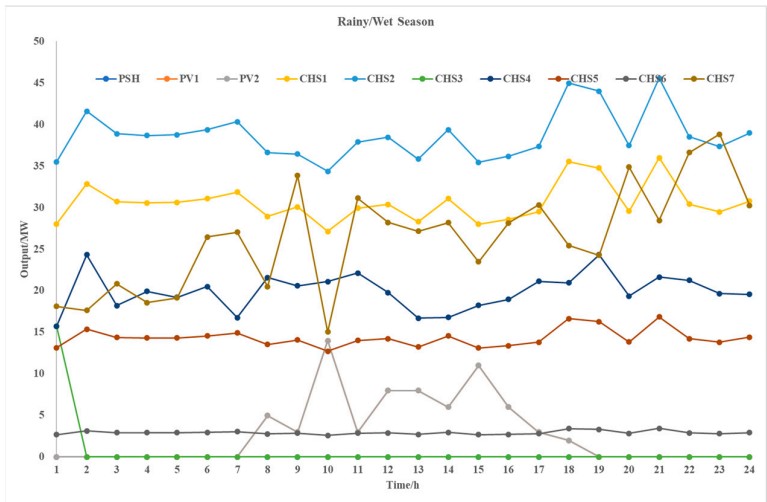

**Figure 6.** Optimal dispatch result in rainy/wet season scenario.

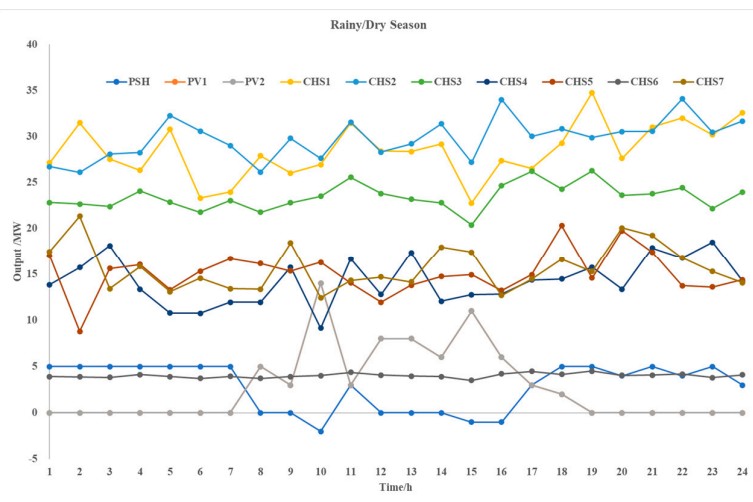

**Figure 7.** Optimal dispatch result in rainy/dry season scenario.

In Figure 4, since the local load is dominated by living load with gentle fluctuation, and the cascaded hydropower system is in heavy output state with high-level PV output

superimposed in sunny/wet season scenario, a considerable part of water energy needs to be stored in the cascaded reservoir, and the PSH system starts the pumping mode for energy storage at the same time. The power output from the natural runoff flow of the cascaded system plays a decisive role in this scenario.

It can be seen in Figure 5 that in the sunny/dry season scenario, the natural runoff flow of the cascaded system reduced sharply. In this case, if the PV system is without power output, it needs the cascaded reservoir to increase discharge flow, and the PSH station needs to start the generation mode to compensate the comprehensive power output. If the PV output is in peak hours, the cascaded hydropower output is weakened, and the cascaded reservoir and PSH station start to store energy according to the real-time load demand. The power output in this scenario is mainly based on the PV output and the compensating output of the cascaded system.

In the rainy/wet season scenario, PV stations generally have a weak power output, as shown in Figure 6. The power output from natural runoff flow of the cascaded system coordinating the regulating output from cascaded reservoir is mainly used to balance the load demand.

As shown in Figure 7, in the rainy/dry season scenario, the cascaded reservoir increases the discharge flow, and the PSH system starts the power generation mode to jointly compensate for the power shortage caused by the sharp reduction of the natural runoff flow of the cascaded system. The supplementary power output from the cascaded hydropower reservoir, combined with the output of PSH system, is taken as the main body in this scenario.

In order to test the effectiveness of the data-driven DRO-coordinated dispatch method for the CHPP system proposed in this paper, the comparative analyses from three aspects are carried out according to the results of the application example.

### 4.1. The Influence of Data Size

The confidence coefficients $\alpha_1$ and $\alpha_\infty$ are set to 0.2 and 0.8, respectively, and calculate the unit average generation cost covering day-ahead dispatching cost and real-time regulation cost with different data size. The calculation results are shown in Table 3.

**Table 3.** The unit average cost with different data size.

| Data Size/pcs | Average Cost/¥/MWh |
|---|---|
| 100 | 221.0963 |
| 500 | 144.7245 |
| 1000 | 138.7216 |
| 2000 | 125.6966 |
| 5000 | 122.9082 |
| 10,000 | 109.9088 |

It can be seen from Table 3 that with the increase of the data size, the unit average cost is gradually decreases. The reason for this situation is that the increase of data size reduces the deviation of its probability distribution and makes it closer to the true distribution. This will decrease the conservativeness of the DRO problem. In addition, when the data size exceeds 1000, the decline of average cost slows, and it can be concluded that data size 1000 can ensure that the optimal solving process meets the corresponding economic and robustness requirements.

### 4.2. The Influence of Confidence Coefficient

Coefficient $\alpha_1$ is set to 0.30, 0.60, and 0.90, and $\alpha_\infty$ is set to 0.50, 0.75, and 0.99, respectively; the configuration of the rest of the parameters remains unchanged. In this case, the calculation for the unit average generation cost of the CHPP system covering the real-time regulation cost and day-ahead dispatch cost of the different confidence combinations in the sunny/wet season scenario is carried out, the results of which are depicted in Table 4.

Table 4 reveals that the unit average cost gradually rises with an increase in the confidence combination value. This is obvious to understand, because the high confidence level indicates more uncertainties covered by the optimization problem. In coordinating the dispatch of the complementary power generation system, a greater need for flexible generation resources arises to handle the growing uncertainties, resulting in an increase in the unit average generation cost. The influence trend of different confidence levels on the unit average cost in Table 4 has confirmed the validity of the comprehensive norm constraint and its linearization in the proposed DRO-coordinated dispatching method.

**Table 4.** The unit average cost with different confidences.

| Types/¥/MWh | $\alpha_\infty = 0.50$ | $\alpha_\infty = 0.75$ | $\alpha_\infty = 0.99$ |
|---|---|---|---|
| $\alpha_1 = 0.30$ | 138.4412 | 138.9613 | 139.1146 |
| $\alpha_1 = 0.60$ | 138.8901 | 139.0761 | 139.2721 |
| $\alpha_1 = 0.90$ | 139.1626 | 139.4247 | 139.4837 |

For the comparison between the comprehensive norm and single 1-norm or ∞-norm constraint, $\alpha_1$ is set to 0.30, 0.60, and 0.90, respectively, and $\alpha_\infty$ is set to 0.99 to form the norm constraint. The unit cost under each norm constraint is shown in the fourth column of Table 4. When only considering ∞-norm constraint ($\alpha_\infty = 0.99$), the unit average cost is ¥139.7633/MWh, which is larger than values in the fourth column. It means that due to double norm taken into account, the conservativeness of the comprehensive norm constraint has been improved. Similarly, $\alpha_1$ is set to 0.90, and $\alpha_\infty$ is set to 0.50, 0.75, and 0.99, respectively to form the norm constraint. The unit average cost under each formed comprehensive constraint is shown in the fourth row of Table 4. When only considering 1-norm constraint ($\alpha_1 = 0.90$), the unit average cost is ¥140.5411/MWh, which is also larger than the values in the fourth row. It also means that the conservativeness of the comprehensive norm constraint has been improved due to the double norm considered.

*4.3. Comparative Study*

To validate the accuracy and efficiency of the suggested coordinated dispatch method, comparative analysis among SO, RO, and probabilistic chronologic production simulation proposed in [39] and our study is carried out.

Taking sunny/wet season scenario as an example, the main parameters of the DRO-coordinated dispatching method for CHPP $\alpha_1$, $\alpha_\infty$, $K$, and data scale are set to 0.20, 0.80, 50, and 1000, respectively. The comparison results between DRO, SO, and traditional RO are depicted in Table 5. As depicted in Table 5, the outcome of DRO falls between the results obtained from the other two methods, which confirms that the coordinated dispatch method has a better robustness. Compared with traditional RO, the DRO method shows a better economy. In conclusion, the DRO-coordinated dispatch method presented in this paper strikes a harmonious equilibrium between economic considerations and robustness.

**Table 5.** Comparison results of DRO, SO, and traditional RO.

| Types | DRO | SO | Traditional RO |
|---|---|---|---|
| Unit Average Cost /¥/MWh | 138.7216 | 137.9051 | 191.3382 |

For the comparison between the DRO and probabilistic chronologic production simulation method, the unit average cost of the two methods and the calculation time is listed in Table 6. It can be seen that the maximum cost obtained by the DRO is smaller than that of the production simulation method, and the average cost of the two is not much different. This is because the results obtained by the probabilistic production simulation method are closely related to the data scale, and the results fluctuate greatly, while the DRO has better

stability. For another aspect, the calculation results of the two methods are close, and the calculating efficiency of the data-driven DRO-coordinated dispatching method is slightly better than that of the probabilistic chronologic production simulation method. The time comparison in Table 6 verifies the accuracy and efficiency of the DRO method when it is used in the CHPP-complementary-coordinated dispatch.

**Table 6.** Results and calculation time of the two dispatch methods.

| Types | DRO | Probabilistic Chronologic Production Simulation |
|---|---|---|
| Unit Maximum Cost/¥/MWh | 146.8864 | 151.2611 |
| Unit Average Cost/¥/MWh | 138.7216 | 138.6383 |
| Calculation Time/s | 58.3 | 71.6 |

## 5. Conclusions

The CHPP complementary generation system's multi-dimensional uncertainty and intricate coupling characteristics have made it extremely challenging to optimize coordinated dispatching. The outputs of typical optimization techniques like SO and RO are always conservative and inefficient, and most of the time it is difficult to obtain a practical dispatch strategy that satisfies the CHPP's optimal dispatching requirements. Therefore, this paper introduced a data-driven DRO technology and proposed a two-stage coordinated dispatching method for the CHPP complementary system. The main conclusions of this paper are as follows:

(1) A data-driven DRO two-stage dispatching model was established to meet the dispatching requirements of CHPP system, and an efficient C&CG algorithm was introduced to divide the model into MP and SP for iterative cycling calculation.

(2) The conservativeness of the DRO dispatching model is inversely proportional to data size, and proportional to the comprehensive norm confidence level. It is further verified that DRO algorithm organically integrates the advantages of SO and RO, and the algorithm achieves a better robustness (Unit Average Cost: ¥138.7216/MWh) than SO (¥137.9051/MWh), and a better economy than RO (¥191.3382/MWh).

(3) The DRO-coordinated dispatching method for the CHPP system can realize the dispatching performance of the probabilistic chronologic production simulation method while improving calculation efficiency. In the same calculation scale, the DRO can save 18.6% calculating time, and achieved a higher efficiency.

Future research will focus on maximizing the power generation income of each subject in the complementary combined power generation system, fully motivating each subject to participate in the complementary power generation, and based on this, obtain a joint optimal scheduling scheme for the CHPP system.

**Author Contributions:** Conceptualization, S.Z. and G.Q.; methodology, S.Z.; validation, S.Z., G.Q. and Y.L.; formal analysis, S.Z.; investigation, S.Z.; resources, L.D. and Y.S.; data curation, S.Z. and L.D.; writing—original draft preparation, S.Z. and G.Q.; writing—review and editing, Y.L.; visualization, Y.S.; supervision, G.Q. and Y.L.; project administration, L.D.; funding acquisition, G.Q. All authors have read and agreed to the published version of the manuscript.

**Funding:** This research was funded by National Nature Science Foundation of China grant number 52307124, the Fundamental Research Funds for the Central Universities grant number YJ2021162, and the Science and Technology Department of Sichuan Province grant number 2021LDTD0016.

**Data Availability Statement:** Data are available on request due to restrictions on privacy or ethics. The data presented in this study are available on request from the corresponding author.

**Conflicts of Interest:** Authors Shuai Zhang, Lijie Ding, and Yue Shui were employed by the State Grid Chengdu Power Supply Company, the State Grid Sichuan Electric Power Research Institute, and the Skill Training Center of Sichuan Electric Power Corporation of State Grid, respectively. The

remaining authors declare that the research was conducted in the absence of any commercial or financial relationships that could be construed as a potential conflict of interest.

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
