# Peer review of "Data-Driven Distributionally Robust Optimization-Based Coordinated Dispatching for Cascaded Hydro-PV-PSH Combined System"

_electronics, doi:10.3390/electronics13030667_

Round 1

Reviewer 1 Report

Comments and Suggestions for Authors This is a very good paper. With an interesting topic and good structure. It could be accepted after being modified during a "minor revision". Here are the specific comments: 1. The literature review could be increased by adding more works. 2. The topic importance should be added as the motivation in the beginning of the introduction and also, abstract. 3. What suggestion could be drawn for future research works? 4. The reference for all the equations should be given. 5. Increase the discussion part by adding more explanations. 6. The paper should be checked once again carefully and completely for English editing. Comments on the Quality of English Language

See the report

Author Response

We sincerely thank you for your careful review. Please find the enclosed attachments for the detailed response letter.

Reviewer 2 Report

Comments and Suggestions for Authors

Generation and storage is a very important and topical problem. The work of many authors focuses on solving this problem. The reviewed paper proposes a dispatching method for a combined hydro-PV-pumped storage cascade system. The proposed method has been tested on a 24-node test system available in the literature. The paper is well structured. The theoretical aspects are presented in a logical form. The content of the paper is of interest to a wide range of electricity professionals. In my opinion, the paper presented is interesting and valuable, however, it needs several corrections:

1. Incorrect numbering of the figures (figure 1 appears twice).

2. The schematic of the test system (24-nodes) is not presented.

3. Too poor commentary on the results presented in Tables 1 and 2.

4. Conclusions too general. I suggest more quantitative data to support the posted conclusions.

Author Response

(The authors gave the same response as above.)

Reviewer 3 Report

Comments and Suggestions for Authors

Paper presents an interesting and comprehensive study, but some improvements such as follows are recommended:

- in lines 64-67, please revise "Different from RO, the chance-constrained method does not guarantee that the decision satisfies any value within the variable boundary [19], Its physical meaning is clear and easy to accept, but the optimization model is generally non-convex and difficult to solve for random variables with complex probability distributions [20]."; meaning is unclear;

- in line 83, "...with SO and Ro, DRO does not need..."; is it Ro or RO?;

- a paper reminder could be introduced in Introduction section;

- in eq. 1 please revise "C dcychead" and "Pi,t H×0" in eq. 3;

- in line 352 please revise "patching model is shown in Error! Reference source not found";

- please revise figures numbering (eg. "Figure 1. Structure and operational mode of a CHPP hybrid power system." in line 123 and "Figure 1. Calculating process of the two-stage DRO model" in line 354;

- for a better readability please illustrate "The improved 24-node test system in [38]...";

- quality of figures 3-6 should be significantly improved for a better readability;

- Conclusion section should be extended and main authors contributions should be properly highlighted. 

Comments on the Quality of English Language

Minor misspelling and mistyping errors should be revised.

Author Response

(The authors gave the same response as above.)

Reviewer 4 Report

Comments and Suggestions for Authors

Dear Authors,

Your manuscript is praiseworthy for its original thinking and in-depth analysis of the difficulties facing CHPP systems. The robustness of your suggested solution is shown by the clarity with which you provide the data-driven complete norm constraints and by the comparison with conventional techniques. I sincerely appreciate your efforts and am excited about the possible effects of your work in the sector. To improve the quality of your work I suggest you address the below comments.

·        In line 11 what do you mean by multi-renewable and their multiple uncertainties if you mean variable renewable energy sources (Solar and Wind) then be clear and state it clearly because other renewable energy sources are not associated with uncertainty.

·        Specify the term "MP-SP framework" upon first use or consider providing a brief explanation to enhance clarity.

·        It would be helpful to briefly mention the key findings or contributions in the abstract.

·        To grab the reader's attention right away, consider including a summary of the manuscript's innovations or contributions in the introduction.

·        Consider providing a brief explanation or motivation for the choice of the 1-norm and ∞-norm in the data-driven DRO algorithm.

·        The presentation of the mathematical expressions is coherent. However, to improve accessibility for readers who might not be specialists in optimisation or DRO, think about including a succinct, understandable explanation or example.

·        Think about including a brief justification of CCG's advantages in the context of the CHPP system and why it is preferred over alternative approaches in the part on the Solution Algorithm for the DRO Dispatch Model.

·        Consider emphasizing the practical implications of the research and potential avenues for future work.

Regards

Comments on the Quality of English Language

Minor English editing is required.

Author Response

(The authors gave the same response as above.)

Round 2

Reviewer 1 Report

Comments and Suggestions for Authors

The revision has been done in a satisfactory way by taking all the comments into consideration. Therefore, the current version could be recommended for publication in the journal.

Reviewer 4 Report

Comments and Suggestions for Authors

Dear Authors,

Your endeavors in addressing the comment are appreciable; there are no further comments from my side, and I recommend it for publication.

Regards,